# Practical Adversarial Attacks on Brain–Computer Interfaces

## Abstract

Deep learning has been widely employed in brain–computer interfaces (BCIs) to decode a subject's intentions based on recorded brain activities enabling direct interaction with computers and machines. BCI systems play a crucial role in motor rehabilitation and have recently experienced a significant market boost as consumer-grade products. Recent studies have shown that deep learning-based BCIs are vulnerable to adversarial attacks. Failures in such systems might cause medical misdiagnoses, physical harm, and financial damages, hence it is of utmost importance to analyze and understand in-depth, potential malicious attacks to develop countermeasures. In this work, we present the first study that analyzes and models adversarial attacks based on physical domain constraints in EEG-based BCIs. Specifically, we assess the robustness of EEGNet which is the current state-of-the-art network for embedded BCIs. We propose new methods to induce denial-of-service attacks and incorporate domain-specific insights and constraints to accomplish two key goals: (i) create smooth adversarial attacks that are physiologically plausible; (ii) consider the realistic case where the attack happens at the origin of the signal acquisition and it propagates on the human head. Our results show that EEGNet is significantly vulnerable to adversarial attacks with an attack success rate of more than 50%. With our work, we want to raise awareness and incentivize future developments of proper countermeasures.

## 1 Introduction

Recent work has shown that adversarial perturbations can cause state-of-the-art (SoA) deep learning models to misbehave in various domains including vision (Szegedy et al., 2014; Goodfellow et al., 2015), NLP (Li et al., 2019a; Zhang et al., 2020), speech (Qin et al., 2019; Li et al., 2019b), and biomedicine (Finlayson et al., 2019; Han et al., 2020). Neural networks have been applied in brain–computer interfaces (BCIs) achieving impressive results (Lawhern et al., 2018; Dose et al., 2018). A BCI enables direct interactions with external devices based on brain activities, typically recorded using electroencephalographic (EEG) systems. It can provide a communication pathway for severely paralyzed patients or assist in rehabilitation (Chaudhary et al., 2016). Besides medical applications, recent developments in wearable devices have pushed BCIs towards consumer-grade products to improve life quality (Aricò et al., 2020), e.g., the Interaxon Muse headband for stress relief (Arsalan et al., 2019) or the Emotiv headset for controlling drones (Marin et al., 2020) and ground vehicles (Zhuang et al., 2021). Safety in BCI systems is paramount (Dutta, 2020; Bernal et al., 2021), because a failure would cause misdiagnoses, user frustration, or even danger while driving a wheelchair or controlling a drone, causing physical and financial damages.

Zhang & Wu (2019) were the first to show that EEG-based BCIs are vulnerable to adversarial attacks by proposing an unsupervised fast gradient sign method (FGSM) (Goodfellow et al., 2015). More recent work has proposed a more practical attack where a universal adversarial perturbation (UAP) is computed once and can be applied to all EEG trials without learning it for every new input (Liu et al., 2021). Both works assume that the acquired signals are sent to a remote compute engine, e.g., a computer, and the attacker can alter the signals during the transmission by attaching a "jamming" module between the signal preprocessing step and the classifier. Recent developments in smart edge computing (Akmandor & Jha, 2018; Beach et al., 2021) eliminate the need for data transmission, making this attack scenario inapplicable. Novel BCI solutions (Kartsch et al., 2019; Wang et al., 2020) embed the signal processing and classification directly at the sensor edge. A more practical

adversarial example has been identified by Meng et al. (2021). It consists of a square-shaped signal that can be added to EEG trials before the preprocessing step. However, the attack is proposed as a backdoor key, which means that the attacker has direct access to the training dataset and pollutes it with adversarial examples, which is improbable if the attacker is not directly involved in the data acquisition or in the training of the classifier. Li et al. (2019b) have shown an attack scenario in the audio domain by considering the on-board edge processing of a wake-word detection system, where an adversarial audio trace is delivered to the environment causing denial-of-service (DoS). No similar studies can be currently found in the BCI domain.

**Challenges: Designing natural attacks and modeling its propagation.** Unlike in audio applications where the signal can simply propagate over-the-air and is sensed by a microphone, extra modeling is required to evaluate the signal propagation in BCIs based on the physical properties of the biological tissues. In this work, rather than assuming a "jamming" module between the preprocessing and the classification steps as in related works, we consider a more realistic and practically applicable attack scenario where the adversarial perturbations are introduced at the *source* of the data acquisition, as showcased in Figure 6 in Appendix A. This can be achieved, for example, via electromagnetic waves delivered to the environment (Dutta, 2020) or via transcranial current stimulation with electrical current delivered directly to the scalp (Bodranghien et al., 2017; Fertonani et al., 2015), by exploiting wearable devices, such as smart glasses or over-ear headsets (Flowneuroscience, 2021; Marin et al., 2020). The adversarial perturbations translate into electrical signals propagating over the scalp and are sensed by the electrodes in addition to the EEG signals.

To guarantee the imperceptibility of the attacks, previous works in BCIs create perturbations that are small in amplitude (Zhang & Wu, 2019; Jiang et al., 2019; Liu et al., 2021), limiting the attack success rate (ASR). Increased perturbation's amplitude yields higher ASR (Meng et al., 2021), but makes the attack more easily detectable. Moreover, the generated perturbations are square-shaped, which is implausible for biosignals. Han et al. (2020) are the first to observe square-wave artifacts in biosignals' attacks and propose smooth perturbations for electrocardiograms (ECGs). No similar works have been found for EEGs.

**This work: Practical attacks on BCI models.** To address the above technical challenges, and for analyzing the vulnerability of embedded BCI models in practical scenarios, we design a new attack algorithm that generates smooth adversarial examples based on the signals' first derivative and model its propagation over the scalp based on a realistic head model by taking into consideration the attack source and the electrical and physical properties of the conducting tissues. This enables the creation of practically effective perturbations, that can be delivered by an external device to attack EEG-based BCIs at the source of signal acquisition. We attack the most energy-efficient network that has been embedded on microcontrollers for smart wearable BCIs called EEGNet (Lawhern et al., 2018; Schneider et al., 2020). It is a resource-friendly convolutional neural network (CNN) and is the SoA in terms of accuracy and energy-efficiency trade-off (Belwafi et al., 2018; Malekmohammadi et al., 2019; Wang et al., 2020; Schneider et al., 2020).

We evaluate our methods and show experimental results on BCIs based on the motor imagery (MI) paradigm, which is of special interest among others because it can be asynchronously self-paced without external stimuli (Freer & Yang, 2020). By imagining the movement of different body parts, the decoded intention is translated into control signals. It is widely applied in several BCI applications, such as the control of wheelchairss (Yu et al., 2018), prosthetic armss (Elstob & Secco, 2016), ground vehicles (Zhuang et al., 2021), and in communication (Brumberg et al., 2016). It has been proven to be the most difficult task to be attacked among the most common BCI paradigms (Zhang & Wu, 2019; Meng et al., 2021). We evaluate our methods by "fooling" the victim model to always predict "rest" class. This essentially yields a DoS attack, because resting-state EEG signals are generally interpreted as no subject's intention decoded, i.e., no control action needs to be taken by the BCI system (Yu et al., 2018). While for healthy subjects it might solely cause user frustration and financial losses, for severely paralyzed patients it can lead to loss of communication and independence. We generalize our methodology to an other MI task of BCI Competition IV-2a dataset and believe that it can be easily adapted to other BCI paradigms.

**Main contributions.** Our main contributions are:

- We design a new method to generate smooth adversarial perturbations that are physiologically plausible and imperceptible to the human eye.

- We consider a practical scenario where the perturbation is added at the signal acquisition source and model its propagation constrained by the physical properties of the human scalp.
- The first study of adversarial perturbations in BCI to consider the practical scenario of smart edge computing and physical signal propagation. We create both local and global perturbations and show that our attacks consistently achieve a success rate of $> 50\%$ in different settings pointing to the significant vulnerability of the SoA embedded EEGNet.

We hope that our work raises awareness for potential risks and motivates the future development of appropriate countermeasures.

## 2 BACKGROUND

### 2.1 CLASSIFICATION IN BCIS

We first describe the commonly used approach in BCIs for classification, consisting of a preprocessing step and a classifier. The brain activity is recorded with an EEG device which samples $N_{ch}$ channels at rate $F_s$. We define one trial $j$ as $(\mathbf{X}^{(j)}, y^{(j)})$, where $y^{(j)} \in \{0, 1, ..., N_{cl} - 1\}$ is the true label of $N_{cl}$ MI tasks, and $\mathbf{X}^{(j)} \in \mathbb{R}^{N_s \times N_{ch}}$ the multi-channel recording defined as

$$\mathbf{X}^{(j)} := \left( \mathbf{x}_0^{(j)}, \mathbf{x}_1^{(j)}, ..., \mathbf{x}_{N_{ch}-1}^{(j)} \right), \tag{1}$$

with $\mathbf{x}_i^{(j)} \in \mathbb{R}^{N_s}$ corresponding to the recording of the $j$-th trial and the $i$-th channel containing $N_s$ temporal samples. For simplicity, we denote $\mathbf{X} := \mathbf{X}^{(j)}$ and $y := y^{(j)}$.

The EEG recordings are often preprocessed with a band-pass filter, e.g., using a Fast Fourier Transform (FFT) filter $h_{bp}(\cdot)$, before being fed to a classifier, yielding

$$\mathbf{X}_{bp} = H_{bp}(\mathbf{X}) = (h_{bp}(\mathbf{x}_0), h_{bp}(\mathbf{x}_1), ..., h_{bp}(\mathbf{x}_{N_{ch}-1})). \tag{2}$$

Finally, the preprocessed signal $\mathbf{X}_{bp}$ is classified with a trainable model $f$ and is mapped to $\mathbf{p} := f(\mathbf{X}_{bp})$, where $\mathbf{p} \in \mathbb{R}^{N_{cl}}$ contains the output probabilities, e.g., originating from a softmax activation as final operation in $f$. The model's final prediction $\hat{y}$ is the index with the maximum score in $\mathbf{p}$:

$$\hat{y} = \hat{f}(\mathbf{X}_{bp}) = \underset{y \in \{0, ..., N_{cl}-1\}}{\operatorname{argmax}} f(\mathbf{X}_{bp})[y]. \tag{3}$$

### 2.2 INSTANCE-BASED ATTACKS

Instance-based attacks try to fool an EEG classifier $f$ to misclassify an EEG signal $\mathbf{X}$ to a targeted class $y_t$. In this section, we describe the attack directly on the classifier $f$ without considering the preprocessing $H_{bp}$; the inclusion of the preprocessing is described in Section 3.3. We define an adversarial example as any $\mathbf{X}^* = \mathbf{X} + \mathbf{V} \in \mathbb{R}^{N_s \times N_{ch}}$ such that

$$\hat{f}(\mathbf{X}) \neq \hat{f}(\mathbf{X}^*) = y_t. \tag{4}$$

**FGSM.** The FGSM (Goodfellow et al., 2015) generates an adversarial perturbation $\mathbf{V} \in \mathbb{R}^{N_s \times N_{ch}}$ of magnitude $\epsilon$ which points in the negative direction of a loss function's gradient:

$$\mathbf{V} = -\epsilon \cdot \operatorname{sign}(\nabla_\mathbf{X} \cdot l(\mathbf{X}, y_t)), \tag{5}$$

where the loss function contains the negative log likelihood

$$l(\mathbf{X}, y_t) = -\log(\mathbf{p}[y_t]) = -\log(f(\mathbf{X})[y_t]). \tag{6}$$

As $\mathbf{p}$ is the output of the softmax activation function, equation 6 becomes a cross-entropy loss which maximizes the output $\mathbf{p}[y_t]$ while minimizing the remaining outputs.

**PGD.** The projected gradient descent (PGD) (Madry et al., 2018) is a variant of the basic iterative method (Kurakin et al.), generally considered to be more effective than FGSM. PGD aims to find a perturbation by iteratively taking small steps of size $\alpha$ in the gradient's direction and projecting the resulting perturbation back to the sample's neighborhood after each iteration. We randomly initialize the attack inside the $L_\infty$ ball of radius $\epsilon$ and update the attack $\mathbf{V}_{t+1}$ for any iteration $t$ with

$$\mathbf{V}_{t+1} = \operatorname{clip}_\epsilon(\mathbf{V}_t - \alpha \cdot \operatorname{sign}(\nabla_\mathbf{V} l(\mathbf{X} + \mathbf{V}_t, y_t))), \tag{7}$$

where $\alpha$ is a step size smaller than $\epsilon$ which decays linearly with each iteration and the function $\operatorname{clip}_\epsilon(\cdot)$ clips the signal at the maximum desired amplitude $\epsilon$.

## 2.3 Universal attacks

UAPs have been introduced by Moosavi-Dezfooli et al. (2017) in the context of natural images, seeking to find an image-agnostic perturbation that fools the classifier on any input image. In the BCI domain (Liu et al., 2021), we seek to find a perturbation $\mathbf{V} \in \mathbb{R}^{N_s \times N_{ch}}$ such that

$$\hat{f}(\mathbf{X} + \mathbf{V}) \neq \hat{f}(\mathbf{X}) \text{ for "most" } \mathbf{X} \sim D, \tag{8}$$

where $D$ is the distribution of the EEG data. The UAP can be determined by optimizing the negative log-likelihood loss with respect to $\mathbf{V}$ using batch gradient descent on the trials in the training set.

## 3 Modeling practical attacks in BCI

This section is the main contribution of the paper: we present a design of practical DoS attacks on MI-BCIs that operates at the source of the signal acquisition. We propose a new method to eliminate the square wave artifacts to generate adversarial examples that are natural and physiologically plausible. The perturbation is emitted by a smart, adversarial device placed close to the ear, e.g., a smart glass or in-ear headphones, and is propagated to the individual EEG electrodes over the scalp's skin. As can be experimentally observed on measured EEG traces (Merlet et al., 2013; Sazgar & Young, 2019), the same electrical source, e.g., electrocardiographic activities, is sensed by each EEG electrode with different degrees of attenuation and delay. We present a practical propagation model that determines the magnitude and delay for every individual electrode based on the distance along the scalp to the adversarial device. The perturbation is trained end-to-end to fool the classifier to always output "rest," hence DoS, while respecting the spatial model and the amplitude constraints to remain imperceptible.

### 3.1 Design and Assessment of Physiologically Plausible Attacks

PGD-designed attacks on EEG tend to form perturbation signals which resemble a square-wave artifact (see Figure 2), an effect that has been observed on ECG data, too (Han et al., 2020). However, EEG signals are of random nature and can be modeled as frequency dependent stationary or non-stationary random processes (Karlekar & Gupta, 2014). To this end, we introduce a new loss term in the PGD optimization such that the perturbation resembles the random nature of EEG signals, which we achieve by promoting signal changes represented in the first order derivative. We estimate the per-channel derivative $\mathbf{V}' = (\mathbf{v}'_0, \mathbf{v}'_1, ..., \mathbf{v}'_{N_{ch}-1}) \in \mathbb{R}^{N_s-1 \times N_{ch}}$ using the sample-wise difference:

$$\mathbf{v}'_c[t] := \mathbf{v}_c[t] - \mathbf{v}_c[t-1] \quad t \in \{1, 2, ..., N_s - 1\}, c \in \{0, 1, ..., N_{ch} - 1\} \tag{9}$$

The additive loss term is determined by $l_1(\mathbf{V}) = -\frac{\beta}{\epsilon} \sum_{c=1}^{N_{ch}} ||\mathbf{v}'_c||_1$, where $|| \cdot ||_1$ is the $\ell_1$-norm, $\epsilon$ the maximum perturbation amplitude, and $\beta \geq 0$ a weighting factor. When designing a one-dimensional perturbation, the derivative loss becomes $l_1(\mathbf{v}) = -\frac{\beta}{\epsilon} ||\mathbf{v}'||$.

**Measuring the Plausibility of Attacks**  None of the previous works have given quantitative measures to assess the physiological plausibility of an EEG adversarial attack. In this work, we propose data-driven measures for quantifying the naturalism of an attack. We compute either the cross correlation, the Euclidian distance, or the cosine similarity between the attacked signal and the original EEG, and average the values over the $N_{ch}$ channels and over the samples in the dataset.

### 3.2 Spatial propagation model

So far, a perturbation signal was designed for every individual channel. It is unrealistic for an attacker to perturb the signal for all individual channels simultaneously; hence, we consider a more practical use case where the perturbation signal $\mathbf{v} \in \mathbb{R}^{N_s}$ is emitted from one location, e.g., from an adversarial device placed on the left side of the subject or close to the left ear. More specifically, in this study, we assume that the EEG electrode at the position T9 according to the international 10-10 system (Sch), which is the closest to the left ear, senses the largest perturbation. The signal subsequently propagates over the skin to each electrode, which results in an individual magnitude and delay depending on the distance between the adversarial device and the electrode. More formally, we model the sensed perturbation at channel $i$ and time instant $t$ as

$$h_i(\mathbf{v}, \lambda_m, \lambda_d)(t) := m(l_i, \lambda_m) \cdot \mathbf{v}(t - d(l_i, \lambda_d)), \tag{10}$$

where $m(l_i, \lambda_m)$ and $d(l_i, \lambda_d)$ are the magnitude and the delay respectively, both of which depend on the distance $l_i$ and on characteristic parameters $\lambda_m$ and $\lambda_d$. We define the resulting multi-channel perturbation $\mathbf{V} \in \mathbb{R}^{N_s \times N_{ch}}$, which is added to the multi-channel EEG signal, as

$$\mathbf{V}(\lambda_m, \lambda_d) = H(\mathbf{v}, \lambda_m, \lambda_d) := (h_0(\mathbf{v}, \lambda_m, \lambda_d), h_1(\mathbf{v}, \lambda_m, \lambda_d), ..., h_{N_{ch}-1}(\mathbf{v}, \lambda_m, \lambda_d)). \quad (11)$$

We estimate the distance $l_i$ between the electrode at position T9 and the remaining, attacked positions using the 10-10 system and a head model with a radius of 8.7 cm (Algazi et al., 2001). We decouple the distance-dependent modeling of the magnitude and delay, explained in the following paragraphs.

**Magnitude.** For modeling the magnitude, we assume that the adversarial device injects or induces a current $I$, yielding a potential $V$ measured near T9. The current propagates over the head surface through the skin to each of the remaining attacked electrodes, which can be modeled as a cylindrical resistor with resistance

$$R_i = \frac{l_i}{\sigma A}, \quad (12)$$

where $\sigma$ is the conductivity of the skin which can be in the range of [0.28, 0.87] Siemens/m (Vorwerk et al., 2019), and $A$ is the area of the skin conductor. The potential at electrode $i$ is $V_i = V - I \cdot R_i$, and hence the magnitude can be described as

$$m(l_i, \lambda_m) = 1 - \frac{V - V_i}{V} = 1 - \frac{I}{V\sigma A}l_i = 1 - \lambda_m l_i, \quad (13)$$

where we further constrain $0 \leq m(l_i, \lambda_m) \leq 1$. The characteristic magnitude parameter $\lambda_m$ represents the complex interplay between input current, voltage, conductivity, and area, covering various attack scenarios. We consider different characteristic magnitude parameters $\lambda_m \in [1, 15]$. A large $\lambda_m$ represents cases with large attenuation and limited propagation, i.e., a limited set of neighboring electrodes sense the perturbation. Conversely, a small $\lambda_m$ covers cases with lower attenuation where the perturbation can propagate further and infects all electrodes. We consider also an intermediate case where around half of the electrodes are affected by the attack with $\lambda_m = 5$. Appendix B provides examples of the magnitude of the spatial propagation on the head model.

**Delay.** The propagation of a signal on the head surface yields a position-dependent phase angle or delay, as shown by experimental measurements of related studies (Plutchik & Hirsch, 1963; Qiao et al., 1994). The delay stems from a combination of resistive and capacitive components that are encountered during the propagation of the signal, which can be modeled as an RC-circuit with resistance $R$, capacity $C$, and time constant $\tau = R \cdot C$ that relates to the group delay. Specifically, the contacts between the electrodes and the skin are predominantly capacitive whereas the skin itself is both resistive and capacitive (Kim et al., 2010). As explained in the previous part, an increasing distance between the attacker and the target electrode yields a larger resistance $R$. As a result, the time constant $\tau$ and the delay increase too.

Here, we model a linear distance-delay relation. We rely on a study by Plutchik & Hirsch (1963), which conducted human skin impedance and phase angle measurements by placing electrodes at an approximate distance of 10 cm and applying voltages with frequencies in the range 2–1000 Hz. When assuming a linear frequency-phase relation in low-frequency region (Qiao et al., 1994), one can derive the group delay to be 2.8 ms when considering a measured angle of $10°$ at 10 Hz. As those measurements were conducted for only one distance, we extrapolate the delay for the remaining distances using a rectified linear model:

$$\lambda_d \cdot (l_i - l_0) > 0 \; ? \; d(l_i, \lambda_d) = \lambda_d \cdot (l_i - l_0) + d_0 : d(l_i, \lambda_d) = 0, \quad (14)$$

where $d_0 = 2.8$ ms is the delay at distance $l_0 = 10$ cm. The delay depends not only on the distance, but also on other parameters such as the electrode-to-skin contact, the humidity of the skin, etc. To this end, we evaluate the propagation of the attack with different characteristic delay parameters $\lambda_d \in [0.1, 0.563]$ s/m. With $\lambda_d = 0.1$ we cover the cases where very little delay happens, while the largest considered $\lambda_d = 0.563$ s/m yields a maximum delay of 0.1 s at the farthest electrode T10, which is in alignment with the observed EEG measurements (Merlet et al., 2013; Sazgar & Young, 2019). Similarly to $\lambda_m$, we showcase also for an intermediate value of $\lambda_d = 0.3$ which corresponds to a delay of 0.053 ms at T10.

---

**Algorithm 1:** Generation of physiologically plausible UAP.

---

**input** : $\mathbf{X}_{train}$, EEG training samples; $\lambda_m, \lambda_d$, spatial propagation parameters; $\beta$, weight of derivative loss term; $\epsilon$, maximum perturbation amplitude; $G$, number of PGD iterations; $E$, number of epochs

**output** : $\mathbf{v}$, adversarial perturbation

---

1   $\mathbf{v} \leftarrow \mathcal{U}(-\epsilon, \epsilon) \in \mathbb{R}^{N_s}$;                     `// Initialisation`
2   **for** $e \leftarrow 1$ **to** $E$ **do**
3     Shuffle $\mathbf{X}_{train}$;
4     **for** *each batch* $\mathbf{B} \in \mathbf{X}_{train}$ **do**
5        $\alpha \leftarrow \frac{\epsilon}{2}$;
6        **for** $g \leftarrow 1$ **to** $G$ **do**
7           $\mathbf{V} \leftarrow H(\mathbf{v}, \lambda_m, \lambda_d)$;                `// Spatial propagation`
8           $\mathbf{p} \leftarrow f(H_{bp}(\mathbf{B} + \mathbf{V}))$;         `// Model pass with perturbation`
9           $\mathbf{v} \leftarrow \mathbf{v} - \alpha \cdot \text{sign}\left(\nabla_{\mathbf{v}}\left(l(\mathbf{p}, y_{rest}) - \frac{\beta}{\epsilon}||\mathbf{v}'||_1\right)\right)$;    `// Update w/derivative`
10          $\mathbf{v} \leftarrow \text{clip}_\epsilon(\mathbf{v})$;                      `// PGD projection`
11          $\alpha \leftarrow \frac{0.1 - \frac{\epsilon}{2}}{G} \cdot g + \frac{\epsilon}{2}$;             `// Learning rate update`
12        **end**
13     **end**
14 **end**

---

### 3.3 ATTACK DESIGN

We present practical DoS attacks in BCIs that respect domain constraints such as maximum amplitude, spectral distribution, physiological plausibility, and the spatial propagation of the perturbation. To this end, we formulate a general objective function that contains the spatial propagation, the preprocessing step, and the first order derivative loss term:

$$\mathcal{L}_{tot}(\mathbf{X}, \mathbf{v}, \lambda_m, \lambda_d) = l\left(H_{bp}(\mathbf{X} + H(\mathbf{v}, \lambda_m, \lambda_d)), y_{rest}\right) - \frac{\beta}{\epsilon}||\mathbf{v}'||_1, \tag{15}$$

where $l(\cdot, \cdot)$ is the negative log-likelihood loss defined in equation 6 and $\beta$=1e-6 is a scalar that weights the contribution of the derivative loss term. We compare different attack scenarios:

**Instance-based attacks.** A perturbation is computed using either FGSM or PGD based on the knowledge of the currently attacked EEG signal $\mathbf{X}$. FGSM computes the perturbation as stated in equation 5, where the $\epsilon$ defines the perturbation amplitude which is varied between 1–50 mV. Alternatively, we compute the perturbation using PGD with $G$=10 iterations, where each iteration consists of a gradient-based update of the perturbation and a projection to the $L_\infty$ ball with radius $\epsilon$ (see equation 7). The update rate $\alpha$ is initialized with $\epsilon/2$ and linearly decreased with each iteration, reaching a final value of 0.1 mV at iteration 10. The PGD computation is restarted 5 times with different initial perturbations, which are drawn from a uniform distribution within the range $[-\epsilon, +\epsilon]$.

**Universal attacks.** A universal perturbation is computed for all the samples in the training data. We optimize the UAP objective function

$$\min_{\mathbf{v}} E_{\mathbf{X} \sim D} \mathcal{L}_{tot}(\mathbf{X}, \mathbf{v}, \lambda_m, \lambda_d) \quad \text{s.t.} ||\mathbf{v}||_\infty \leq \epsilon \tag{16}$$

using batched PGD. We pass a batch of 16 samples together with the current perturbation through the preprocessing and classifier, compute the loss function, and update the perturbation based on the negative gradient with consecutive projection to the $L_\infty$ ball with radius $\epsilon$. This step is repeated $G$=10 times before processing the next batch. Overall, the UAP is learned for $E$=10 epochs.

**Propagation model.** We distinguish between three use cases of spatial propagation model, where in all cases either an instance-specific attack or a universal attack can be computed: Case 1) Ignore the propagation model: a multi-channel perturbation $\mathbf{V}$ is computed, which attacks each channel individually, replacing the terms $H(\mathbf{v}, \lambda_m, \lambda_d)$ by $\mathbf{V}$ and $\mathbf{v}'$ by $\mathbf{V}'$ in equation 15. Case 2) Consider

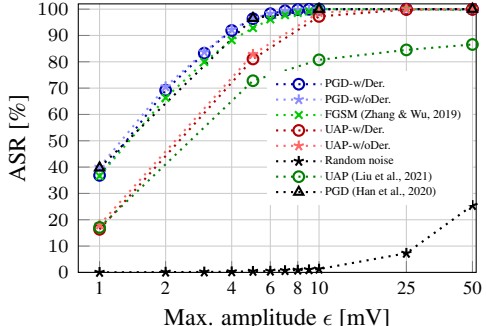

**Figure 1:** Performance of random noise, FGSM, PGD, and UAP with and without derivative loss term.

| | $\eta$ [$10^{-3}$V$^2$] | | | $\ell_2$-norm [mV] | | | $\gamma$ [%] | | |
|---|---|---|---|---|---|---|---|---|---|
| $\varepsilon$[mV] | (a) | (b) | (c) | (a) | (b) | (c) | (a) | (b) | (c) |
| 1 | 3.31 | 1.98 | 3.42 | 20.8 | 15.2 | 21.2 | 99.89 | 99.93 | 99.89 |
| 5 | 16.6 | 7.76 | 17.3 | 99.1 | 61.2 | 102 | 97.99 | 99.22 | 97.87 |
| 10 | 32.5 | 12.6 | 34.1 | 191 | 112 | 198 | 93.82 | 97.47 | 93.49 |
| 25 | 74.9 | 27.3 | 79.2 | 461 | 263 | 475 | 79.61 | 90.05 | 78.92 |
| 50 | 125 | 39.0 | 135 | 823 | 462 | 855 | 64.17 | 79.70 | 63.06 |

**Table 1:** Plausibility metrics for PGD attack (a) without derivative term, (b) with the derivative loss term, and (c) with a Gaussian kernel (Han et al., 2020). The smaller the cross correlation $\eta$ and the Euclidian distance $\ell_2$-norm, and the higher the cosine similarity $\gamma$, the more natural the generated attack.

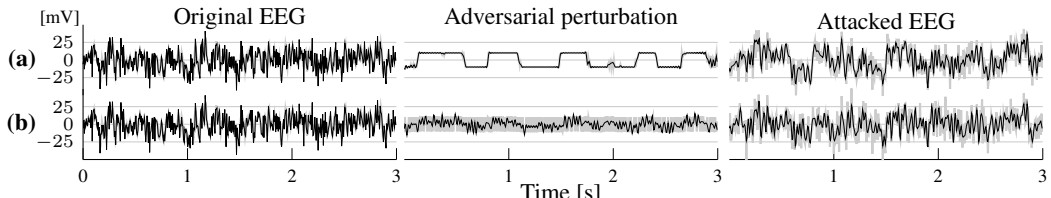

**Figure 2:** A successful attack (a) without and (b) with derivative loss term (PGD, $\epsilon$=10mV). The background traces show the original signal before the preprocessing filter.

the propagation model: a single-channel perturbation $\mathbf{v}$ is computed and tested with a specific propagation configuration $\lambda_m$ and $\lambda_d$. Case 3) Consider a use-case where the attacker does not know the spatial propagation model and computes the same perturbation $\mathbf{v}$ for all channels. The actual propagation model is applied during testing to model the real-world signal propagation.

**End-to-end algorithm.** We illustrate the algorithmic procedure for designing a physiologically plausible UAP in Algorithm 1. Analogously, the proposed methods of derivative loss term and model propagation are applied with PGD. The hyperparameters $\alpha$, $\beta$, the number of PGD iterations and the restarts, the batch size and the number of epochs in UAP are determined based on a cross-validated grid search on the training set.

## 4 EXPERIMENTS AND RESULTS

We evaluate our methods on the Physionet EEG Motor Movement/Imagery Dataset (Goldberger et al., 2000; Sch) tackling inter-subject challenges, and generalize to subject-specific inter-session dataset IV-2a of BCI Competition (Brunner et al., 2008) (See Appendix C).

**Dataset.** The Physionet dataset contains valid EEG recordings of 105 subjects (Dose et al., 2018) and is publicly available under Open Data Commons Attribution License v1.0. We use the MI recordings that contain tasks of the imagination of left against right fist for 3 s. The EEG trials were recorded with $N_{ch}$=64 channels sampled at $F_s$=160 Hz, yielding $N_s$=3·160=480 samples per trial. Additional baseline runs provide resting-state data, where the subjects did not perform any tasks while having eyes open. Overall, we get a total of 6615 trials with $N_{cl}$=3 balanced classes "left", "right", and "rest."

**Training and validation.** We train and validate both the classification models and the generated adversarial examples with a 5-fold cross-validation, splitting the dataset into 84 subjects used for training and 21 subjects used for validation to effectively test the model on inter-subject variability. Similar to Wang et al. (2020), which achieved SoA performance on this dataset, the baseline model is trained for 100 epochs using Adam with $\beta_1$=0.9, $\beta_2$=0.999, and batch size of 16. The learning rate is 0.01 and decreased by a factor of 10 at epochs 20 and 50, achieving an average accuracy of 74.78%.

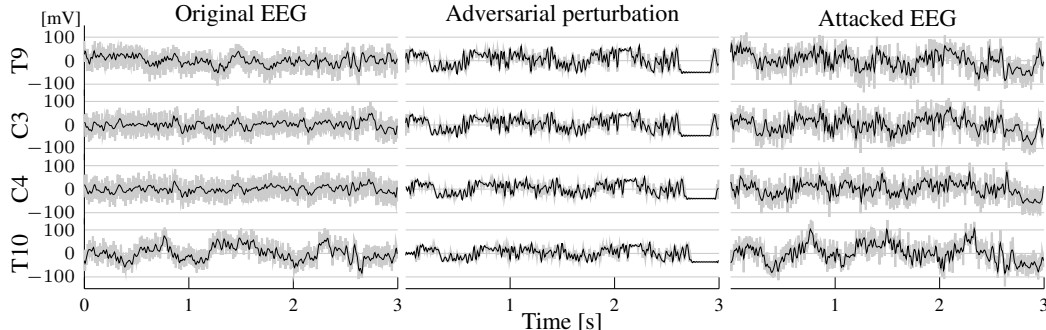

**Figure 3:** A successful attack with derivative loss term and spatial constraints $\lambda_m = 1$ and $\lambda_d = 0.563$ (PGD, $\epsilon$=50mV). The background traces show the original signal before the preprocessing filter.

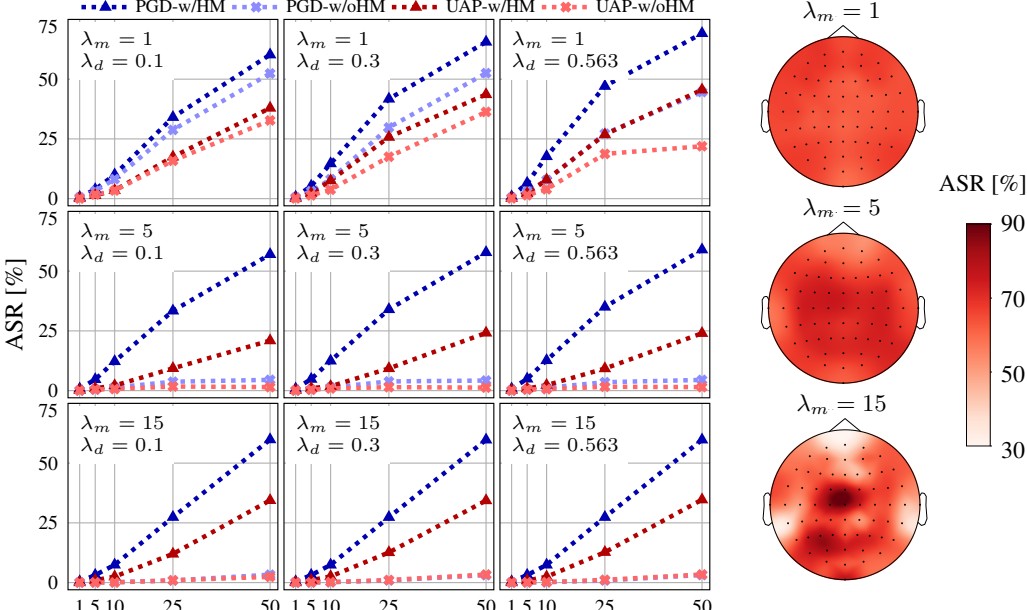

**Figure 4:** ASR of PGD and UAP in Case 2), i.e., computed with head model (w/HM), and in Case 3), i.e., computed without head model (w/oHM).

**Figure 5:** ASR with the PGD attack propagating from different EEG channels with fixed $\lambda_d$=0.3 and variable $\lambda_m$.

An FFT band-pass filter $h_{bp}$ with a customary passband of 0.1–40 Hz (Lawhern et al., 2018) is used as preprocessing step in both baseline and attack experiments. To determine the ASR, we compute the ratio between the successfully fooled trials, i.e., trials now classified as "rest", and the total number of attacked trials, where we only consider the ones initially correctly classified as "left"/"right".

**Physiologically plausible attacks.** We first analyze the instance-based attacks without considering the propagation model (Case 1), depicted in Figure 1. We compare our methods against random noise with amplitude $\epsilon$ as in (Zhang & Wu, 2019), FGSM that is the same as in (Zhang & Wu, 2019) with targeted scenario, and a UAP designed specifically for EEG (Liu et al., 2021). For both FGSM and PGD, the ASR increases together with the maximum amplitude $\epsilon$ of the perturbation. They always outperform the random noise, with PGD performing slightly better than FGSM. They reach the maximum ASR of 99.97% with 10mV. The post-attack classification accuracy drops from 74.78% to 48% for a perturbation amplitude of 2 mV and to 33% for 10 mV and higher amplitudes. Figure 2a shows the signals of a successful attack using PGD. The adversarial perturbation has a square-wave form which negatively affects the natural shape of the EEG signal. By adding the proposed derivative term, the square-wave artifacts are significantly reduced (2b), making the perturbation more

physiologically plausible. When comparing the power spectral density of the original and attacked signals, the attacked signal designed without derivative presents large components in low frequencies, making it more easily detectable. Whereas the attack with derivative loss better resembles the power spectral density of the original signal (see Appendix D). Moreover, the introduction of the derivative term does not degrade the ASR (Figure 1). The quantitative measures between the original and the adversarial samples in Table 1 demonstrate that our proposed method with derivative term generates adversarial samples that are more similar to the original EEG, allowing them to remain imperceptible even with high $\epsilon$ (Appendix D). We reproduce the attacks using a Gaussian kernel as in (Han et al., 2020). After tuning the kernel size and variance of the Gaussian kernel, the method could not improve the plausibility metrics. The inferior performance of the Gaussian kernel could stem from the different nature of the signal: it was originally designed for ECGs which have a pseudo-periodic structure.

We extend the application of the derivative term to the UAP attack, while still not considering the propagation model (Case 1). Figure 1 shows a comparison in performance for different values of $\epsilon$. The saturation in ASR is reached with higher $\epsilon$, i.e., 99.94% with 50mV. This is expected since the UAP is a more difficult attack where a single set of perturbations per EEG channel is generated for all the test samples. Likewise in PGD, the ASR does not drop with the addition of the derivative term. We reproduce the UAP proposed by (Liu et al., 2021). Our UAP consistently reaches higher ASR.

**Spatial Propagation.**     Finally, we introduce the spatial constraints in the signal propagation over the scalp (Case 2). We consider 9 different scenarios by combining 3 realistic attenuation configurations $\lambda_m \in \{1, 5, 15\}$ with 3 delay configurations $\lambda_d \in \{0.1, 0.3, 0.563\}$, which capture the range described in Section 3.2. For evaluating the highest achievable attack efficiency, we test a scenario where the attacker is assumed to know the propagation model: the adversarial perturbation is generated and evaluated on fixed spatial parameters $\lambda_m$ and $\lambda_d$, shown in Figure 4, where the ASR reaches up to 69.2% with PGD and 45.6% with UAP at 50mV. Figure 3 depicts an example of a successful attack with the highest perturbation amplitude. The introduction of the spatial constraints makes the attack problem harder yielding seldom square distortions. However, the resulting EEG signals still resemble physiological random processes typical of EEGs. Next, we ablate the spatial constraints during generation and test the resulting perturbations on the 9 above-mentioned scenarios (Case 3). The ASR drops significantly, especially for $\lambda_m$=5 and $\lambda_m$=15 where the attenuation of the perturbation over the scalp is greater (see Figure 7), and with the global UAP attack, where the attacker does not have access to the attacked EEG signals.

Our spatial propagation models allows us to identify the vulnerability of the individual EEG channels. Figure 5 shows the ASR when initiating an attack from a specific channel (T9, T10, etc.) and propagating it to the rest of the head. In the case with the greatest attenuation ($\lambda_m$=15) we find the maximum ASR at the electrode Cz between the regions of the electrodes C3 and C4, which are the most relevant ones for MI of the left and right hand tasks (Pfurtscheller & Lopes da Silva, 1999). We compute the pre- and post-attack confusion matrices for attacks from T9 and T10 (see Appendix E). When the attack propagates from the left side (T9), more samples with ground-truth label "right" can be fooled to "rest", while the attacks from the right side (T10) are more effective "left" labels.

Overall, our methods successfully generates perturbations resembling natural noise in EEGs, that can be added at the source of the signal acquisition and are propagated over the scalp, creating attacked signals that are physiologically plausible. Similar results have been observed on the BCI Competition IV-2a dataset, shown in Appendix C.

## 5    Conclusion

With the incentive of improving security in BCIs, in this work, we demonstrated that DoS attacks are feasible and effective despite physical domain constraints. Experimental results reveal potential risks of realistic attacks on smart wearable BCIs and incentivize the need for future development of defense mechanisms while designing deep learning models to be embedded in smart wearable BCIs. Our detailed analysis on each EEG channel shows that special attention has to be paid, combined with the findings in neuroscience, to the brain regions that are found responsible for a specific task. In future work, the proposed attacks can cover uncertainty in the propagation model and the timing of the MI activity. Moreover, hardware implementations of such attacks can be created to evaluate the proposed methods in real-world, with the ultimate goal of developing effective countermeasures.

ETHICS STATEMENT

The sore goal of this work is to raise awareness of potential adversarial attacks in BCIs and incentivize the development of coutermeasures, especially in the current moment when the BCIs are facing an increasing growth in applications of everyday life. The active development of smart wearable BCIs is introducing a paradigm shift where the processing algorithms are embedded near the data acquisition. While this improves the system security to a certain extend, with this work we have shown that it is not the only and ultimate way to a safe and reliable BCI, since we have shown that BCI systems are vulnerable also to attacks at the signals' source. We hope that our work sheds light on the fact that practical BCI systems are vulnerable, despite the physical constraints, and motivates the design and development of more reliable and robust BCI systems.

REPRODUCIBLITY

A link to a anonymous downloadable source code of this work is submitted as supplementary materials.

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

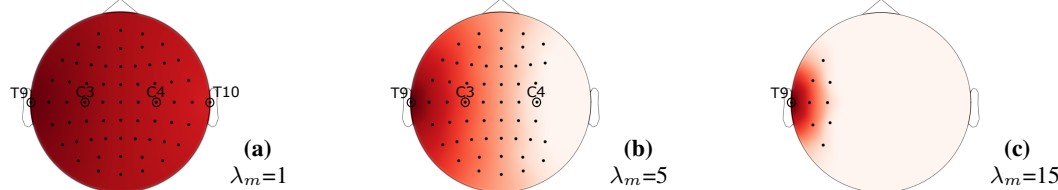

**Figure 6:** Practical adversarial attack scenario in BCIs: a smart device close to the ear emits a perturbation signal which propagates over the head surface to the EEG electrodes.

**(a)**
$\lambda_m=1$

**(b)**
$\lambda_m=5$

**(c)**
$\lambda_m=15$

**Figure 7:** Magnitude of the spatial propagation for different $\lambda_m$. The perturbation is emitted from the left side of the head and propagates over the head surface. The leftmost electrode senses the highest magnitude (red), which linearly decreases towards zero (white) with growing propagation distance and $\lambda_m$. The electrodes which sense the perturbations, i.e., magnitude $>0$, are marked with dots. The electrodes T9, C3, C4, and T10 are labeled for reference.

## A   ATTACK AT THE SOURCE OF SIGNAL ACQUISITION

Fig. 6 illustrates the new attack scenario where the perturbation is delivered to the human scalp and propagates to the sensing electrodes at the source of the signal acquisition.

## B   SPATIAL PROPAGATION MODELS

Fig. 7 illustrates the magnitude of signal propagation using different propagation parameters $\lambda_m = \{1, 5, 15\}$. A large $\lambda_m$ represents cases with large attenuation and limited propagation (e.g., attack over the air) and a small $\lambda_m$ covers cases with lower attenuation where the perturbation can propagate farther (e.g., a smart glass).

## C   EXPERIMENTS ON BCI COMPETITION IV-2A

**Dataset.**   The IV-2a dataset of the BCI Competition contains recordings from nine different subjects and distinguishes between four classes of imagined movements: left and right hand, both feet, and the tongue. 22 different EEG channels were recorded, sampled at 250 Hz. The data was pre-processed with a bandpass filter between 0.1 and 40 Hz. Each subject completed two recording session on two different days, where the first session is used for training and the second for testing as per the rules of the competition. Each session contains 288 trials.

**Training and validation.**   We train a separate baseline model per subject using Adam optimizer with $\beta_1=0.9$ and $\beta_2=0.999$, a batch size of 32, and 500 epochs. The learning rate is 0.001 achieving an average accuracy of 71.79%. This dataset does not contain the rest class. We choose to design an attack that aims to fool the classifier to always predict "tongue." Moreover, we apply a maximum perturbation amplitude of $\epsilon \in [0.01, 10]$ mV due to the lower signal amplitude encountered in this dataset.

**Results.**   Fig. 8 compares the ASR of different attacks without considering the propagation model (Case 1). Generally, a minimal perturbation amplitude of 1 mV and 2 mV suffices to achieve 100% ASR with PGD and UAP, respectively. The addition of the derivative loss term does not give any performance degradation in terms of the ASR. The average post-attack classification accuracy drops

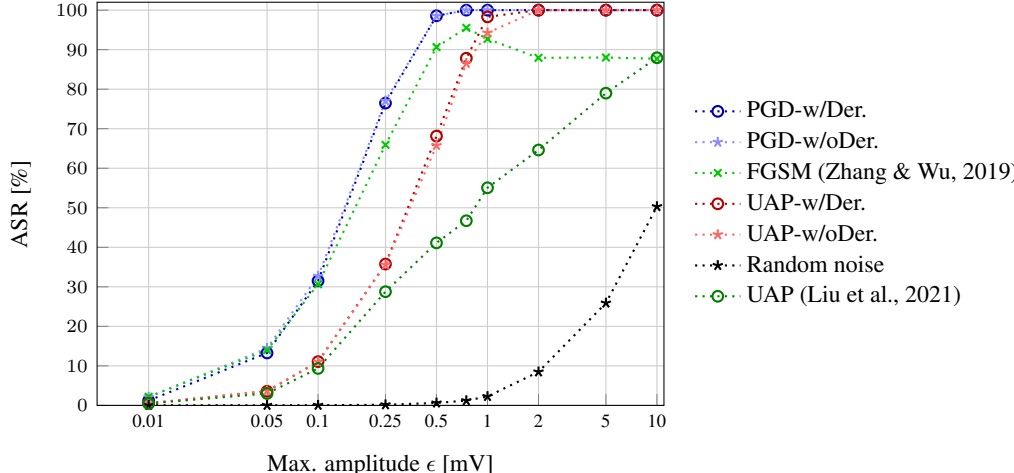

**Figure 8:** ASR on BCI Competition IV-2a with random noise, FGSM, PGD, and UAP with and without derivative loss term.

from 71.79% to 50% for a perturbation amplitude of 0.15 mV and 24.7% for 0.6 mV and higher amplitudes, when PGD with derivative is used.

Fig. 9 shows the ASR for different propagation parameters ($\lambda_m$ and $\lambda_d$) and maximum perturbation amplitudes $\epsilon$. When considering the head model during the design of the attack (Case 2, w/HM), both PGD and UAP reach significantly higher ASR compared to attacks designed without the consideration of the head model (Case 3, w/oHM).

# D  PLAUSIBILITY OF ATTACKS

This section provides power spectral density plots of original signals and attacked signals with and without the derivative loss term, shown in Figure 10. The power spectral density is determined by computing the magnitude squared Fast Fourier Transform of the signals that were illustrated in Figure 2. The attack designed with the derivative loss term has a similar distribution as the original signal, where as the attack without derivative shows large contributions in the low frequency domain (<5 Hz), which were not present in the original signal. These low-frequency components stem from the square-wave shaped attack and can be used as a way to detect the attack; hence, this attack cannot be considered imperceptible.

Moreover, Figure 13 shows the attacks with and without derivative loss term with increasing maximum amplitude $\epsilon$. We can see that for low amplitudes (1 mV and 5 mV) the generated attacks with and without derivative still look like EEGs. At 10 mV, the attack generated without derivative presents minor square-wave artifacts, which could be still imperceptible to a non-expert. With 25 mV and 50 mV, the ones generated without derivative have strong and perceptible square-wave displacements, while the ones generated with our proposed method can still be mistaken as real EEG signals. While with the instance-based attacks, it is not necessary to have more than 10 mV to get a very high ASR (see Figure 1), with the universal attacks and physical constraints, the ASR increases with increasing perturbation amplitude (see Figure 4).

The same observations can be drawn from the plausibility metrics, which have been proposed for the first time in this paper to assess quantitatively the EEG attacks. For example, looking at the cosine similarity ($\gamma$) in Table 1, without the derivative loss term, $\gamma$ drops to 97.99% with $\epsilon = 5$ mV, whereas, with the derivative, $\gamma$ drops to about the same value of 97.47% with $\epsilon = 10$ mV, yielding an increase in ASR from 85% (5 mV) to 99% (10 mV) shown in Figure 1 with PGD.

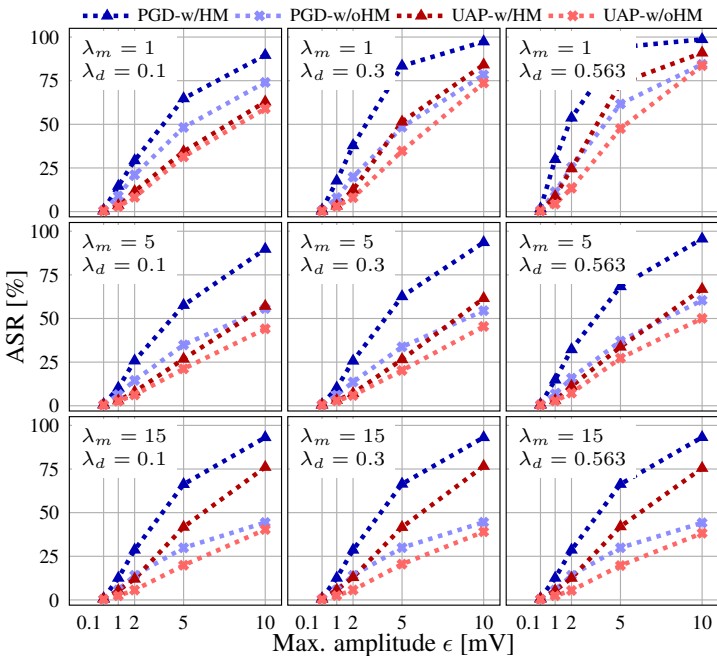

**Figure 9:** Results on BCI Competition IV-2a. ASR of PGD and UAP in Case 2), i.e., computed with head model (w/HM); and in Case 3), i.e., computed without head model (w/oHM).

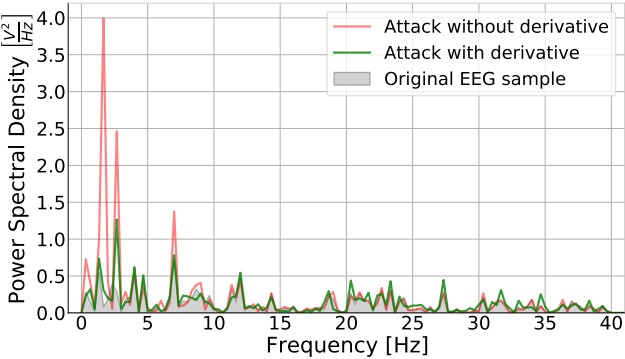

**Figure 10:** Power spectral density comparison of the attack with and without derivative loss term, as well as the original signal shown in Figure 2.

## E  CLASSIFICATION CONFUSION MATRICES

We analyze the confusion matrices before and after the proposed attack. Fig. 11 shows the confusion matrix of EEGNet on the Physionet dataset before the attack, where all classes can be classified with similar accuracy (72.8%–73.5%). Fig. 12 shows the confusion matrices for three different propagation parameters ($\lambda_m \in \{1, 5, 15\}$) and two attack positions (T9 and T10) which correspond to the left and right side of the head. When considering the attacks from the left side, shown in Fig. 12a–12c, more samples with ground-truth label "right" can be fooled to "rest". This is particularly articulated in largely attenuated propagation model ($\lambda_m$=15). In a similar vein, attacks coming from the right side of the head (T10) are more effective on data with ground-truth label "left" (Fig. 12d–12f).

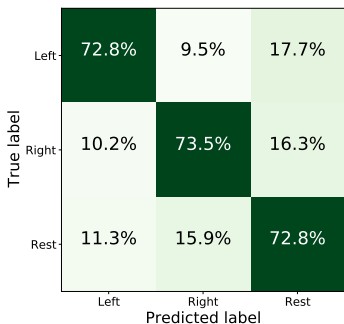

**Figure 11:** Confusion matrix original EEG predictions on Physionet dataset.

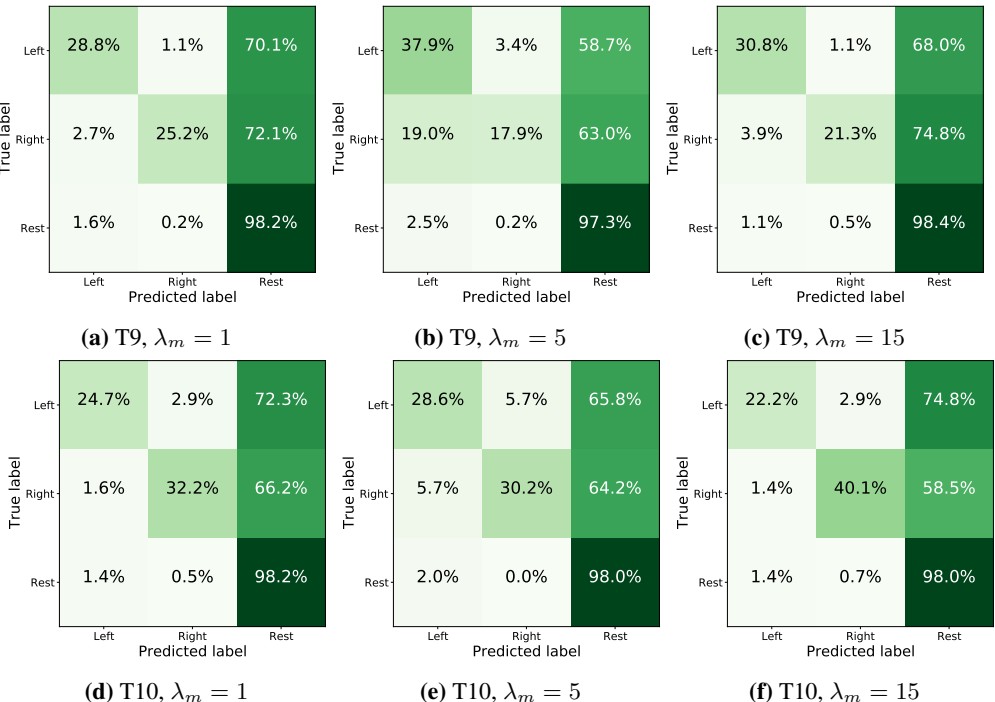

**Figure 12:** Confusion matrices for the Physionet dataset after attacking EEGNet with the proposed PGD attack with derivative and considering the spatial propagation. The attack is either performed from the left electrode (T9) or from the right electrode (T10). We consider different magnitude propagation parameters $\lambda_m$ and a constant delay parameter $\lambda_d$=0.3.

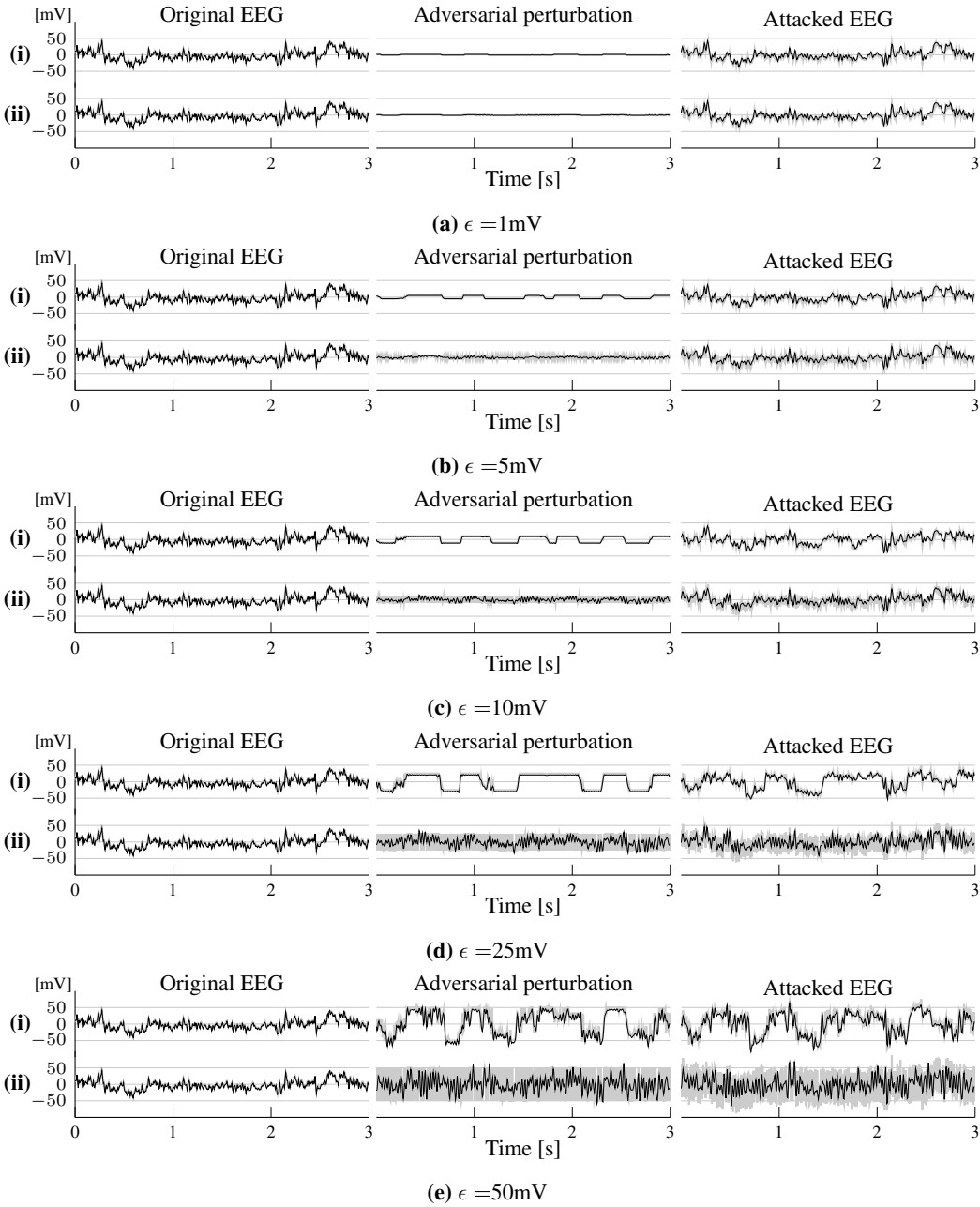

**Figure 13:** A successful PGD attack on Physionet dataset (i) without and (ii) with derivative with different values of maximum amplitude $\epsilon$.

