# OpenReview forum: "Practical Adversarial Attacks on Brain--Computer Interfaces"
_ICLR.cc/2022/Conference — ICLR 2022 Submitted_

### Official Review · Reviewer_n45G · 2021-10-29

**Correctness:** 3
**Technical Novelty And Significance:** 2
**Empirical Novelty And Significance:** 2
**Recommendation:** 3
**Confidence:** 5

**Main Review:**

MI-BCI systems are ultimately motivated to provide alternative communication and control means for people with severe neuromuscular disabilities. In this context, it is not really easy for me to understand the motivation of investigating adversarial attacks to induce DoS for a MI-BCI system in experiments, or at least it is not clearly put in the paper. I believe this motivation can only be applicable to a reasonable extent with certain scenarios like EEG-based biometric identification. Going further, how to possibly overcome this vulnerability that is proposed in the paper is also not discussed or studied, which would be in my opinion a more thorough study if preliminary analyses with standard adversarial training like approaches are at least briefly assessed?

Authors motivate their approach in comparison to existing attacks based on arguments that I am not fully convinced about. For instance, authors argue that the perturbations should be imperceptible and smooth. However, the follow-up argument on realizing artifacts at the source-level relies on e.g., tDCS delivered directly to the scalp (which is generally not imperceptible within the first few milliseconds, when EEG time-series is simply monitored). In fact, every noise-like artifact (sometimes even additive square-waves) can be rather considered imperceptible with EEG, as opposed to the large artifacts caused by tDCS or TMS at the instant. What would the authors comment on that?

Any BCI system that utilizes a discriminative DNN (e.g., EEGNet) is likely to be vulnerable to gradient-based adversarial perturbations as the authors well demonstrate. However most BCI systems (particularly MI-BCIs) that use conventional signal processing methods (i.e., filter-bank common spatial patterns or feature extraction methods based on Riemannian geometry) in fact still yields state-of-the-art (or comparable) performances with respect to deep feature learning models. It would be interesting for the current study to discuss or empirically investigate if such a vulnerability would be present (with similar impact?) in a BCI system running on such conventional feature decoding methods?

Some methodological comparisons are necessary, and missing in the paper. A baseline could simply be to add white noise or square waves to the signals (i.e., random pseudo-perturbations that have a maximum amplitude bound), and obtain a baseline attack success rate. Similarly, comparisons to state-of-the-art ideas on adversarial attacks to BCIs are not included (e.g., [Zhang & Wu, 2019] or [Liu et al, 2021] where the signal attacks/jamming occur after the EEG acquisition phase). For instance, how do the authors' UAP attacks relate to the [Liu et al, 2021] study, when one does not consider the spatial propagation constraints?

Online decoding for MI-BCIs is a very challenging task by itself, especially if one considers inter-subject decoding (as in the paper). Can the authors also present some subject-specific pre-attack and post-attack accuracies to give a better overview of how much the BCI system truthfully suffers from this vulnerability, since pre-attack MI decoding performances can be already lower in the current setting?

Regarding the visualizations in Figure 5, additional power spectral density plots would be helpful to even better demonstrate the similarities between original and attacked EEG signals, as they are generally monitored to check EEG for irregularities.

It would be also interesting to see if the results would look exactly similar, when one imposes the same adversarial attacks with similar parameters, starting from the right ear location electrode T10 which spatially propagates towards T9 the other way around (i.e., impacts C3 less than C4). Accordingly with this question, I would be also curious how are the attack success confusion matrices look like?, i.e., are the attacks having more impact on right hand motor imagery detection due to targeting T9?

Some minor comments:
- Typo in Eq 14.
- End of page 6: "log-loss" -> "negative log-likelihood loss"
- I would label the y-axis of Figure 3 as "attack success accuracy" rather than "accuracy".

**Summary Of The Paper:**

Authors study the vulnerability of brain-computer interface (BCI) systems from a security perspective, and present a physiologically plausible adversarial attack approach on BCI systems that can induce denial-of-service after being transmitted by external devices. Empirical assessments are performed on an offline three-class motor imagery (MI) BCI dataset (right hand, left hand, and rest classes), where denial-of-service indicates a "rest" classification regardless of the subjects' motor imagery intent. Proposed methods are motivated to be physiologically plausible and effective to an extent that can cause failure modes for DNN-calibrated BCI systems.

**Summary Of The Review:**

The paper is well written with a clear narrative, and used methods are well described. Main limitations of this work is on the presentation of the general motivating perspective, as well as a general lack of methodological comparisons/evaluations and discussions on possible countermeasures that one can consider for DoS attacks on BCI systems. Some experiments/comparisons are highlighted and suggested in my main review for the authors.

---

> ### Author Response · Authors · 2021-11-19
> **Response to Reviewer n45G [Part 1 of 2]**
>
> We thank the reviewer for their valuable comments. Please find below our response:
>
> Q1:
> MI-BCI systems are ultimately motivated to provide alternative communication and control means for people with severe neuromuscular disabilities. In this context, it is not really easy for me to understand the motivation of investigating adversarial attacks to induce DoS for a MI-BCI system in experiments, or at least it is not clearly put in the paper. I believe this motivation can only be applicable to a reasonable extent with certain scenarios like EEG-based biometric identification.
>
> R1:
> Resting-state EEGs are interpreted as an idle state. For a patient with severe disabilities, DoS means that the patient is not anymore able to communicate or to control the wheelchair or a prosthetic arm. This causes the loss of the alternative communication pathway or the loss of independence.
> We reworded the introduction to better explain this idea and stated that the proposed methodology can be adapted to “fool” the victim model to other classes, depending on the desired misbehavior of the BCI system.
>
>
> Q2:
> Going further, how to possibly overcome this vulnerability that is proposed in the paper is also not discussed or studied, which would be in my opinion a more thorough study if preliminary analyses with standard adversarial training like approaches are at least briefly assessed?
>
> R2:
> We would like to thank the reviewer for this constructive suggestion. The goal of this study was to create imperceptible and realistic attacks based on physical modeling. We agree with the reviewer that adversarial training could be used to defend against our attacks and consider it as an interesting item for future work.
>
> Q3:
> Authors motivate their approach in comparison to existing attacks based on arguments that I am not fully convinced about. For instance, authors argue that the perturbations should be imperceptible and smooth. However, the follow-up argument on realizing artifacts at the source-level relies on e.g., tDCS delivered directly to the scalp (which is generally not imperceptible within the first few milliseconds, when EEG time-series is simply monitored). In fact, every noise-like artifact (sometimes even additive square-waves) can be rather considered imperceptible with EEG, as opposed to the large artifacts caused by tDCS or TMS at the instant. What would the authors comment on that?
>
> R3:
> The transcranial current stimulation is provided as an example that it is feasible to deliver perturbation as an addition to the EEG signals before the sensing electrodes. Fertonani et al. (2015) have found that the induced sensations can be modulated and it depends on many factors (e.g., direct or alternate current, intensity of the current, size of the stimulating electrodes, etc.). The current stimulus can also be low enough that the subject does not feel anything.
>
> Regarding square waves, as observed by Han et al. (2020), they are not physiologically plausible. Especially when the amplitude of the perturbation is high, they are not imperceptible. We added Figure 13 in the Appendix of the updated submission showing an example with increasing maximum amplitude of the perturbation. We can see that for low amplitudes (1mV and 5mV) the generated attacks with and without derivative still look like EEGs. At 10mV, the attack generated without derivative presents  minor square-wave artifacts, which could be still imperceptible to a non-expert. With 25mV and 50mV, the ones generated without derivatives have strong and perceptible square-wave displacements, while the ones generated with our proposed method can still be mistaken as real EEG signals. While with the instance-based attacks, it is not necessary to have more than 10mV to get a high attack success rate (Figure 1 of the updated submission), with the universal attacks and physical constraints, the attack success rate increases with increasing perturbation amplitude (Figure 4 of the updated submission).
>
> The same observations can be drawn from the plausibility metrics, which have been proposed for the first time in this paper to quantitatively assess the EEG attacks. For example, looking at the cosine similarity (gamma) in Table 1, without the derivative loss term, gamma drops to 97.99% with maximum amplitude epsilon = 5mV, whereas, with the derivative, gamma drops to about the same value of 97.47% with epsilon =10mV, yielding an increase in ASR from 85% (5mV) to 99% (10mV) shown in Figure 1 with PGD.

---

> > ### Author Response · Authors · 2021-11-19
> > **Response to Reviewer n45G [Part 2 of 2]**
> >
> > Q4:
> > Online decoding for MI-BCIs is a very challenging task by itself, especially if one considers inter-subject decoding (as in the paper). Can the authors also present some subject-specific pre-attack and post-attack accuracies to give a better overview of how much the BCI system truthfully suffers from this vulnerability, since pre-attack MI decoding performances can be already lower in the current setting?
> >
> > R4:
> > We conducted additional experiments on the BCI Competition IV-2a dataset that is often used for subject-specific evaluations. We randomly chose the target class as “tongue”, as no rest data has to be classified in this dataset.  Also on this dataset, our proposed attacks outperform the baselines (random noise, Zhang & Wu 2019, Liu et al. 2021) and achieve an ASR of >50% when considering the physical propagation model. The results can be found in Appendix C of the updated submission. We would also be happy to provide results on other targets if needed.
> >
> > The pre-attack accuracy on Physionet dataset is reported in the Training and validation paragraph in Section 4 Experiments and results. We added the post-attack accuracy in the same section of the updated submission. The accuracy drops from 74.78%to 48% for a perturbation amplitude of 2 mV and to 33% for 10 mV and higher amplitudes.
> > The pre- and post-attack accuracy on BCI Competition IV-2a dataset is added to Appendix C of the updated submission. The classification accuracy drops from 71.79% to 50% for a perturbation amplitude of 0.15 mV and 24.7% for 0.6 mV and higher amplitudes.
> >
> > Q5:
> > Regarding the visualizations in Figure 5, additional power spectral density plots would be helpful to even better demonstrate the similarities between original and attacked EEG signals, as they are generally monitored to check EEG for irregularities.
> >
> > R5:
> > We added power spectral density plots of the original signal as well as attacked signal (with and without the derivative loss term) to Appendix D. The attack designed with our derivative loss term has a similar power spectral density distribution as the original signal, whereas the attack without derivative shows large contributions in the low frequency domain (<5Hz), which can be easily detected. Hence, the attack without the derivative loss term cannot be considered as imperceptible.
> >
> > Q6:
> > It would be also interesting to see if the results would look exactly similar, when one imposes the same adversarial attacks with similar parameters, starting from the right ear location electrode T10 which spatially propagates towards T9 the other way around (i.e., impacts C3 less than C4). Accordingly with this question, I would be also curious how are the attack success confusion matrices look like?, i.e., are the attacks having more impact on right hand motor imagery detection due to targeting T9?
> >
> > R6:
> > In addition to Figure 5, which shows the ASR when the perturbation is injected at any electrode position, we added more detailed results of attacks at T9 and T10 in form of confusion matrices in Appendix E. Overall, a similar ASR is achieved when attacking from T9 or T10. Indeed, based on the confusion matrices we can see that attacks on T9 have a larger impact on right hand MI detection, and attacks on T10 on left hand MI.
> >
> > Q7:
> > Some minor comments:
> >     Typo in Eq 14.
> >     End of page 6: "log-loss" -> "negative log-likelihood loss"
> >     I would label the y-axis of Figure 3 as "attack success accuracy" rather than "accuracy".
> >
> > R7:
> > We implemented the suggested changes on the negative log-likelihood loss and the y-axis of Figure 1 of the updated submission. Equation 14 is an 'if-else' statement; hence, it is correct.
> >
> >
> > References
> >
> > Anna Fertonani, Clarissa Ferrari, and Carlo Miniussi. What do you feel if i apply transcranial electric stimulation? Safety, sensations and secondary induced effects. Clinical Neurophysiology, 126(11): 2181–2188, 2015.
> >
> > Xintian Han, Yuxuan Hu, Luca Foschini, Larry Chinitz, Lior Jankelson, and Rajesh Ranganath. Deep learning models for electrocardiograms are susceptible to adversarial attack. Nature medicine, 26 (3):360–363, 2020.
> >
> > Zihan Liu, Lubin Meng, Xiao Zhang, Weili Fang, and Dongrui Wu. Universal adversarial perturbations for cnn classifiers in eeg-based bcis, 2021.
> >
> > Xiao Zhang and Dongrui Wu. On the vulnerability of CNN classifiers in EEG-based BCIs. IEEE Transactions on Neural Systems and Rehabilitation Engineering, 27(5):814–825, 2019.

---

> > > ### Comment · Reviewer_n45G · 2021-11-22
> > > **Thanks to the authors for the responses**
> > >
> > > Thanks to the authors for their revisions and explanations. I have read the comments carefully in detail, and decided to keep my score as it is. Below are my detailed comments.
> > >
> > > - Authors chose to only present an attack concept as a problem in this paper. Regarding my comment on “possible approaches to overcome the vulnerability (e.g., adversarial training as a defense)”, there were no revisions/discussions and this was left for future work.
> > >
> > > - Regarding my concerns and comments on using tDCS as an imperceptible attack tool, authors’ response was also not convincing on my end. I agree and it is known that EEG can be modulated with such stimulation methods (in ways that may not be very clear), but one would realize that there is an electrical stimulation going on by just looking at the EEG recording (this is due to the large electrical interference occurring at the bio-potential sensors within milliseconds of stimulation). Also, as part of the authors’ proposed concept, it should then be justified how to even realize this such that an electrical stimulation will yield the exact same EEG response that the algorithm derives.
> > >
> > > - The authors did not to take a step further in demonstrating the effectiveness of their approach against different signal processing pipelines. Since the authors chose to only present their story for BCIs running on EEGNet, this naturally restricts contributions. There is no edge device or microcontroller yet in the authors’ experiments, and the presented work explores an algorithmic/conceptual approach to potentially perform adversarial attacks. Therefore, technically it should have been possible that the authors show their approach gives similar results for at least another deep signal processing pipeline (e.g., at least another CNN such as DeepConvNet [Schirrmeister et al., 2017]).
> > >
> > > - I could not see a response to my previous question: "...For instance, how do the authors' UAP attacks relate to the [Liu et al, 2021] study, when one does not consider the spatial propagation constraints?". I also did not understand why the authors did not stick to the same dataset for their subject-specific decoding results in the revisions but changed the dataset there.
> > >
> > > Overall, I decided to keep my score as it is.

---

> > > > ### Author Response · Authors · 2021-11-25
> > > > **Second response to Reviewer n45G**
> > > >
> > > > We would like to thank the reviewer for their feedback.
> > > >
> > > > Regarding adversarial training: we believe it is an important topic worth investigating as a standalone rather than a subpart of another work. Similar to the previous related work mentioned by the reviewer (Liu et al, 2021), in this paper we focus on designing adversarial attacks.
> > > >
> > > > Regarding tDSC: to the best of our knowledge, when the stimulation is small enough, it can be imperceptible on the EEG. We will be glad if the reviewer can kindly suggest related literature discussing the large electrical interference occurring at the bio-potential sensors within milliseconds of stimulation. We recognize that the realization of the specific adversarial attack designed by our algorithm needs further investigation.
> > > >
> > > > Regarding DeepConvNet: it performs slightly worse than ShallowConvNet in terms of classification accuracy, which performs similarly to EEGNet, while requiring orders of magnitude more resources in terms of memory and computation. Since our focus is oriented towards embedded BCI in the field of tinyML, we chose the most energy-efficient model and did not consider DeepConvNet. However, we believe our methods can be easily adapted to other deep learning models.
> > > >
> > > > Regarding comparisons to related works: as mentioned in the general response, besides other baselines, we also reproduced the methods by Liu et al (2021) to compare to our approach. The results are added to Figure 1 and show that our method yields up to 16.5% more attack success rate than (Liu et al., 2021).
> > > >
> > > > Regarding the dataset for subject-specific validation: we chose the IV 2a dataset from the BCI Competition for subject-specific validation due to its popularity for this kind of studies in BCI. Reviewer 2xxQ also requested to validate our methods on another dataset, hence, we chose to do subject-specific studies on BCI Competition IV 2a.

---

### Official Review · Reviewer_2xxQ · 2021-11-02

**Correctness:** 3
**Technical Novelty And Significance:** 2
**Empirical Novelty And Significance:** 3
**Recommendation:** 6
**Confidence:** 4

**Main Review:**

This is an interesting study which addresses AI-based medical applications targeted for edge devices. Related work is well presented and contributions are explained, however the novelty of the work is not clear, as some methods for generating smooth adversarial examples for different electrical body signals already exist. I believe similar concepts as used for example in ECG might be applicable to EEG as well. It would be beneficial to a reader to compare those.

The flow of the work should be also improved for better readability. The related work section could be included in the introduction, as the introduction already contains some comparison with previous studies. Figures placement, ordering, descriptions and references in the next definitely need to be improved.  Below Eq. 9 the loss term is defined in words, while it would be better to use the equation. I believe it will be the same as the loss in Eq. 15(?), so maybe consider restructuring the work to improve its flow. Please also explain how attenuation configurations were selected, specifically explain the reason for choosing specific magnitude and delay values.

The presented experiments were performed using one EEG dataset for a specific task of the motor imagery. It would be interesting to verify if the method can scale to other tasks as well and generalize to other datasets. Please elaborate on that.

Also, some other more recent studies on EEG classification using e.g., transformers already exist. What is the reason for examining CNN models? Would transformers be more robust to adversarial attacks in this application, given the fact that they were proved more robust in other studies?



**Summary Of The Paper:**

The paper examines vulnerability of the brain computer interfaces to adversarial attacks. A new method is proposed to generate more realistic, smoother representations of the adversarial attack. A case where an attack happens at a specific location (close to an ear) is studied as a representative case of the perturbation added to the signal acquisition source.

**Summary Of The Review:**

 This is an interesting study, but in my opinion requires some additional work to make the contribution more solid, specifically comparison with other smooth adversarial attacks on electrical signals should be provided, and selection of CNNs vs transformers should be explained. It would be also beneficial to verify the proposed methods with other datasets/other EEG-based tasks if possible.

---

> ### Author Response · Authors · 2021-11-19
> **Response to Reviewer 2xxQ**
>
> We thank the reviewer for the constructive feedback. Please find our response below:
>
> Q1: some methods for generating smooth adversarial examples for different electrical body signals already exist. I believe similar concepts as used for example in ECG might be applicable to EEG as well. It would be beneficial to a reader to compare those.
>
> R1: We reproduced the Gaussian smoothing kernel by Han et al. (2020) and applied it to the EEG signals. The results are added to Table 1 and show that the attacks trained with our proposed derivative loss are significantly more similar to the original EEG signals, i.e., more physiologically plausible for EEGs, than the ones designed with the method proposed by Han et al. (2020).
>
> We could not find any other related works in designing physiologically plausible attacks on biosignals. We will be happy to compare to other relevant works with reproducible codes if the reviewer will kindly suggest.
>
>
> Q2: The flow of the work should be also improved for better readability. The related work section could be included in the introduction, as the introduction already contains some comparison with previous studies.
>
> R2: We would like to thank the reviewer for the valuable suggestion. We now included the related works section in the introduction.
>
>
> Q3: Figures placement, ordering, descriptions and references in the next definitely need to be improved.
>
> R3: We carefully checked and improved the figures’ placement, ordering, descriptions, and references.
>
>
> Q4: Below Eq. 9 the loss term is defined in words, while it would be better to use the equation. I believe it will be the same as the loss in Eq. 15(?), so maybe consider restructuring the work to improve its flow.
>
> R4: We would like to thank the reviewer for the constructive comment. We added the formulation of the additive loss term under Eq. 9. Indeed it is a term that is included in Eq. 15.
>
>
> Q5: Please also explain how attenuation configurations were selected, specifically explain the reason for choosing specific magnitude and delay values.
>
> R5: As described in Section 3.2 of the updated submission, the characteristic parameters are selected based on the models and justified with observations in related works. We correlate the magnitude parameter to the skin conductivity which is selected based on measurements conducted in related works (Vorwerk et al. 2019). We select three magnitude parameters to cover three scenarios (exemplified in Figure 7 in Appendix A):
> - When the attenuation is high, i.e., only a limited set of neighboring electrodes sense the perturbation;
> - When attenuation is low, i.e., all electrodes sense the perturbation;
> - An intermediate scenario where half of the electrodes are attacked.
> Similar arguments apply to the selection of the delay values, based on the EEG measurements in related works (Merlet et al., 2013; Sazgar & Young, 2019). We edited Section 3 to better explain these configurations.
>
>
> Q6: The presented experiments were performed using one EEG dataset for a specific task of the motor imagery. It would be interesting to verify if the method can scale to other tasks as well and generalize to other datasets. Please elaborate on that.
>
> R6: Following the reviewer’s recommendation, we ran our methods on an additional dataset. We selected the IV-2a dataset from the popular BCI Competition (Brunner et al., 2008) following the comment by Reviewer n45G to address the inter-subject variability. We added the additional experiments on the IV-2a dataset in Appendix C. The additional experimental results show that the proposed attack is effective on this dataset too, achieving an attack success rate of over 50%.
>
>
> References
>
> C. Brunner et al. BCI competition 2008 - Graz data set A, 2008. URL http://bnci-horizon-2020.eu/database/data-sets. doi:10.1007/BF00994018.
>
> X. Han et al. Deep learning models for electrocardiograms are susceptible to adversarial attack. Nature medicine, 26 (3):360–363, 2020.
>
> I. Merlet et al. From oscillatory transcranial current stimulation to scalp eeg changes: A biophysical and physiological modeling study. PLOS ONE, 8:1–12, 02 2013.
>
> M. Sazgar and M. G. Young. EEG Artifacts, pp. 149–162. Springer International Publishing, Cham, 2019.
>
> J. Vorwerk et al. Influence of head tissue conductivity uncertainties on eeg dipole reconstruction. Frontiers in neuroscience, 13:531, 2019.

---

> > ### Comment · Reviewer_2xxQ · 2021-11-24
> > **Thank you very much for all the updates.**
> >
> > I think the work has been improved with the introduced changes for other smooth adversarial examples and additional experiments for different datasets. I agree with some other reviews stating that the work could be further enhanced by performing practical experiments and verifying different neural networks, but in my opinion the performed analysis is already quite extensive and I'm ok with adding these additional studies in future work. Thus, I'm willing to change my score to marginally above the acceptance threshold.

---

> > > ### Author Response · Authors · 2021-11-25
> > > **Thank you very much**
> > >
> > > We would like to thank the reviewer for their constructive feedback and their positive assessment of our work.

---

### Official Review · Reviewer_D9vr · 2021-11-02

**Correctness:** 3
**Technical Novelty And Significance:** 3
**Empirical Novelty And Significance:** 3
**Recommendation:** 8
**Confidence:** 4

**Details Of Ethics Concerns:**

The paper presents the details of an adversarial attack which in principle could be harmful for the subjects. Nonetheless this has been explicitly addressed by the authors and is correctly covered in the manuscript

**Main Review:**

I believe this paper addresses a quite relevant topic, of interest for the community: the impact of adversarial perturbations in sensitive wearable systems. I think both motivation and contribution of the work are correct, with a proper statement of the problem and a comprehensive SoA analysis. The paper is well written, with no detecting spelling issues, and overall easy to read.  The overall contribution is adequate, being my only concern related to the completeness of the work and the overall reproducibility of results.

I believe the general structure of the manuscript  is correct. It includes a proper explanation of BCI, SoA, background of the attack, and a dataset that, although limited, given the scarce public dataset for this domain I consider adequate. The part which I consider could be improved, and hence my recommendation, is the description of the method and experimental setup.

Since this approach relies on affecting a ML system to misclasify its input, I’d expect to see a more detailed version of adversarial architecture and it’s training process. Training params are indeed included but after reading the paper several times I’m not sure about the adversarial approach to get access to the gradient of the attacked system.

Honestly I’d rather see less details about the propagation and plausibility of the attack, and even remove some figures (as the Figure1 which do not offer substantial information) and include the end-to-end training algorithm  used to generate the adversarial model (or substitute) in a figure. As I mentioned before I’ve serious doubts that this system could be reproduced  with the details provided in the paper. By the way, I did not see any supplementary material which I understood the authors attached. I’d suggest adding the end-to-end architecture plus training algorithm, with some details and metrics about the training and inference process.

The paper may raise some ethical concerns since the topic is quite sensitive, but I think these matters have been explained adequately by the authors in the manuscript.


**Summary Of The Paper:**

Authors propose a method to perform adversarial attacks on BCI deployed on wearable devices. Using an electrode enclosed in a head mounted device, they show how to propagate a perturbation through the subject’s scalp capable of significantly affecting a EGG based classification model. Authors test the approach on the movements detection domain using a public dataset, showing how the proposed attack can lead to a significant decrease of  the system’s performance.

**Summary Of The Review:**

I consider this paper as an interesting work with a proper motivation and novelty component. The topic addressed (adversarial perturbations on wearable BCI ) can be definitively of interest for the conference audience. The main improvement point in my opinion is the description of method and experimental setup. Further details on the adversarial approach I think would improve the paper. I consider the overall contribution of the work as adequate, and hence my recommendation

---

> ### Author Response · Authors · 2021-11-19
> **Response to Reviewer D9vr**
>
> Thank you for your feedback and the constructive recommendations. Please find our response below:
>
> Q1:
> The part which I consider could be improved, and hence my recommendation, is the description of the method and experimental setup.
> Since this approach relies on affecting a ML system to misclasify its input, I’d expect to see a more detailed version of adversarial architecture and it’s training process. Training params are indeed included but after reading the paper several times I’m not sure about the adversarial approach to get access to the gradient of the attacked system.
>
> R1:
> As per the reviewer’s recommendation, we have added the end-to-end training algorithm used to generate the UAP with our proposed derivative loss term and signal propagation model in Algorithm 1 in Section 3.
>
>
> Q2:
> I did not see any supplementary material which I understood the authors attached. I’d suggest adding the end-to-end architecture plus training algorithm, with some details and metrics about the training and inference process.
>
> R2:
> We have now uploaded anonymous code as supplementary material. We plan on releasing the code upon acceptance.

---

> > ### Comment · Reviewer_D9vr · 2021-11-29
> > **Thanks to the authors for their answer.**
> >
> > First of all thank you to the authors for the update on the paper. I truly believe Figure1 was not neccesary and now the algorithm description is more comprehensive.
> > I still think the reproducibility of the paper is somehow limited and indeed agree with other reviewers about the practical experiments. Those elements would improve the paper for sure but I still think the paper can bring value to the conference. It has a interesting novelty component, I think authors have addressed comments meticulously and overall I believe is good work. Given the scope of the venue I consider the contribution of this paper is adequate. I've no problem in adapting my recommentdation

---

> > > ### Author Response · Authors · 2021-11-30
> > > **Thank you very much**
> > >
> > > We would like to thank the reviewer for their constructive feedback and their positive recommendation.  We are glad that the reviewer finds our contributions sufficiently novel and significant for publication at ICLR.

---

### Official Review · Reviewer_VvTE · 2021-11-04

**Correctness:** 3
**Technical Novelty And Significance:** 2
**Empirical Novelty And Significance:** 2
**Recommendation:** 3
**Confidence:** 5

**Main Review:**

Strengths:
1. the motivation and contributions are well written. There are indeed few studies on the adversarial attacks on BCI.

Weaknesses:
1. This work is NOT actually implementing attack (as implied in the Introduction) with a physical device, but simulating attack! It's a huge difference. Simulation is much more easier and costs much less effort. See more details in the following review summary.

**Summary Of The Paper:**

This work analyzes the adversarial attacks on BCI systems. It induces DoS attack and incorporates domain-specific insights to achieve the adversarial attack. The designed attack is evaluated on a public dataset EEGMMIDB and shows that a popular deep learning model, EEGNet, is vulnerable to adversarial attacks.

**Summary Of The Review:**

The whole paper (Introduction, Fig 1, etc.) implies that this study has a smart device that can be attached to the left ear that will inject well-designed perturbation to collected EEG data. I was excited when reading the paper and I know how hard it is. It requires a multidisciplinary team to design the wireless smart device, manufacture the device and make sure it really works, break the EEG collection equipment (like Emotive headset) which is difficult as the signal transmission is generally encrypted, and so on. I was intended to give an accept (or even a strong accept) as long as they can implement the whole system in the real world.

But it's not the case! The dataset is not locally acquired but is the most well-known public EEGMMIDB dataset, which takes little effort to get. The attack is not really deployed with a smart device at the left ear which can emit perturbations to the source signal, but is simulated. There is no device, no signal transmission, no poisoning attack on source signals.

Without the effort on physical systems and real-world attacks, this paper is still good but not good enough for ICLR. It's a good story, but still a story. So I decided to give a rejection.

---

> ### Author Response · Authors · 2021-11-19
> **Response to Reviewer VvTE**
>
> We would like to thank the reviewer for the valuable feedback. Please find our response below:
>
> Q1:
> This work is NOT actually implementing attack (as implied in the Introduction) with a physical device, but simulating attack! It's a huge difference. Simulation is much more easier and costs much less effort. See more details in the following review summary
>
> R1:
> We agree with the reviewer that real experiments will ultimately demonstrate the effectiveness of the modeling we proposed in this paper. However, this is currently not possible because of ethical considerations and restrictions, as also pointed out by Reviewer D9vr. Further, our main goal is to not harm people, but demonstrate the vulnerability of BCI systems and encourage the development of robust models. In our simulated attacks, we consider the physical properties of the human head, which poses our work closer to a real-world setup than previous related works.
>
> Q2:
> The whole paper (Introduction, Fig 1, etc.) implies that this study has a smart device that can be attached to the left ear that will inject well-designed perturbation to collected EEG data
>
> R2:
> With this work we want to raise awareness for potential risks in smart wearable BCIs in view of the recent active development in smart edge computing, that makes the threat model presented by previous related works inapplicable (Zhang & Wu, 2019, Liu et al., 2021).
> In order to explain this new threat model that we introduce for the first time, we mentioned smart glasses and other smart wearable devices as an example of a possible attacker and Figure 1 of the first submission is to showcase this new attack scenario.
>
> Q3:
> The dataset is not locally acquired but is the most well-known public EEGMMIDB dataset, which takes little effort to get. The attack is not really deployed with a smart device at the left ear which can emit perturbations to the source signal, but is simulated.
>
> R3:
> We did not acquire our own dataset because there exist already many excellent publicly available datasets based on which many works have been published; hence, reproducibility and comparisons are more straightforward. The public EEGMMIDB dataset is chosen for its bigger size.
> As explained above, no physical experiments are performed because of ethical considerations.
>
> Q4:
> Without the effort on physical systems and real-world attacks, this paper is still good but not good enough for ICLR.
>
> R4:
> We thank the reviewer for acknowledging our work as good. We note that several published digital attacks at ICLR including the highly cited PGD attack (Madry et al. at ICLR 2018) were not reproducible in the real world, so we find this criticism of the reviewer harsh.
>
> References
>
> Zihan Liu, Lubin Meng, Xiao Zhang, Weili Fang, and Dongrui Wu. Universal adversarial perturbations for CNN classifiers in EEG-based BCIs, 2021.
>
> Aleksander Madry and Aleksandar Makelov and Ludwig Schmidt and Dimitris Tsipras and Adrian Vladu. Towards Deep Learning Models Resistant to Adversarial Attacks. International Conference on Learning Representations, 2018.
>
> Xiao Zhang and Dongrui Wu. On the vulnerability of CNN classifiers in EEG-based BCIs. IEEE Transactions on Neural Systems and Rehabilitation Engineering, 27(5):814–825, 2019.

---

> > ### Comment · Reviewer_VvTE · 2021-11-19
> > **Thanks for the response**
> >
> > Thank you for the response. I have carefully read the comments and prefer to keep my score not changing. Here is why:
> >
> > - My main question on real-world implementation is not solved (it's not surprising as the implementation is impossible to be done in several days). Frankly, the main challenge preventing real-world application is not ethical considerations but technical difficulties. I have experience in both brain signal analysis and adversarial attacks for a long time, and I have tried to work on attacking BCI systems: 1) apply for ethical permission from the ethics committee is feasible, although it requires a lot of tedious paperwork; 2) the really tough things are on building hardware platform/device and telecommunication, it's even rare to find an open-source sensor (non-commercial, signal transmission is not encrypted) that can precisely collect EEG signals. However, without practical implementation, adopting adversarial attacks on BCI is just a kind of low-hang fruit that has less significance.
> >
> > - Since this work does not contain physical experiments, lots of statements are overclaimed and misleading. For example, 1) the title is ''practical adversarial attacks'' which implies this work is practiced, at least partially practiced, otherwise how you know it's practical. 2) On page 2, one subheading is named "Practical attacks on BCI models", this paragraph has a number of misleading descriptions: " for analyzing the vulnerability ... in practical scenarios", "enables the creation of practically effective perturbations", "can be delivered by an external device to attack EEG-based BCIs", "We attack ...which has been embedded on energy efficient microcontrollers for smart wearable BCIs... "  3) the original Fig 1 is also misleading, I noticed it was moved to Appendix now.
> >
> > - In summary,  I am aware asking for real-world implementation is kind of harsh. However, the overclaim and misleading narration needs significant revision.

---

> > > ### Author Response · Authors · 2021-11-21
> > > **Second response to Reviewer VvTE**
> > >
> > > Thank you for the response. We are glad that the reviewer recognizes the complexity involved in getting the approval needed to even consider real-world experiments and that their views are on a harsher side.
> > >
> > > Regarding the hardware platform/device: openBCI project (https://openbci.com/), born in 2013, aims at providing open-source BCI platforms. They have a GitHub page (https://github.com/OpenBCI) with open-source materials. More specifically for hardware, we can find https://github.com/OpenBCI/V3_Hardware_Design_Files which has been available since 2015.
> > >
> > > Regarding encrypted wireless transmission: as we mentioned in the general response and in the paper, our proposed attack scenario considers cases where the wireless transmission is not necessary. We model attacks that can be applied on the human scalp before the sensing electrodes, hence, there is no necessity of accessing the data during wireless transmission.
> > >
> > > We respectfully disagree that the work we do is a low-hanging fruit as it requires addressing the challenges mentioned in the paper. Further, as mentioned, our motivation is to study the vulnerability of the classifier also in the realistic scenario where the perturbations are not simply added to the EEG signals after the acquisition but is delivered at the source before the sensing electrodes and the signal propagation is constrained by physical and electrical properties of the biological tissues.
> > >
> > > Regarding the statements: 1) we use the standard interpretation of the word “practical” from the Oxford dictionary which states “(of an idea, plan, or method) likely to succeed or be effective in real circumstances; feasible” which is in line with what we show in the paper; 2) in the paragraph under the subheading “Practical attacks on BCI models”, we clearly state that we “model” the propagation over the scalp. The attack can be delivered by an external device to attack EEG-based BCIs “at the source of signal acquisition”, which is one of our main contributions. As mentioned in the general response, we attack EEGNet, because it has been embedded on microcontrollers citing the work by Schneider et al. (2020), and is the most energy-efficient. 3) As we explained in the general response, Fig. 1 of the first submission was an illustration to showcase the more realistic attack scenario that we introduce for the first time in the BCI domain. It has been moved to the Appendix to address the comments by Reviewer 2xxQ.
> > >
> > > Finally, we would like to clarify that it is not our intention to mislead the reader. We are willing to adjust the title and the introduction if the reviewer can kindly suggest an alternative that they consider appropriate to help avoid the confusion and at the same time accurate in describing the actual contributions of our paper.
> > >
> > >
> > > References
> > >
> > > Tibor Schneider, Xiaying Wang, Michael Hersche, Lukas Cavigelli, and Luca Benini. Q-EEGNet: an energy-efficient 8-bit quantized parallel EEGNet implementation for edge motor-imagery brain-machine interfaces. In 2020 IEEE International Conference on Smart Computing (SMARTCOMP), pp. 284–289, 2020.

---

### Author Response · Authors · 2021-11-19
**Major additions and answer to common questions**

We would like to thank the reviewers for their valuable and constructive comments. We first report the major additions we've made and answer to general questions. We then provide detailed responses to each of the reviewers to individually address their concerns.

The major modifications are:

- We reworked the sections of Introduction and Related works to better highlight the motivation and our contributions and to improve the flow of the paper. We reworked the section “Modeling practical attacks in BCIs” to better explain our methodology and algorithm.
- We performed additional experiments to reproduce the related work by Han et al. (2020) which proposed smooth adversarial examples for ECG signals. We computed and added the similarity metrics in Table 1. The results show that our method based on signals’ first derivative generates adversarial examples that are significantly more similar to original EEG signals than the methods by Han et al. (2020), hence, less likely to be detected.
- We added a random noise-based baseline and reproduced the methods by Liu et al (2021) to compare to our approach. The results are added to Figure 1 and show that our method significantly outperforms the random noise in terms of attack success rate (ASR) and yields up to 16.5% more ASR than (Liu et al., 2021). Concerning the work by Zhang & Wu (2019), it corresponds to the FGSM results in the Figure 3 of the first submission. Their main contribution is to propose an unsupervised FGSM (UFGSM) attack that tries to “fool” the classifier to misclassify to a class different than the one predicted on the original output regardless of whether the original signal was correctly classified. In our case, we consider only correctly classified samples so UFGSM=FGSM for all experiments in Figure 1 of the updated submission.
- We performed additional experiments on a new dataset (BCI Competition IV-2a) to generalize our methods and evaluate the subject-specific performance. The additional experimental results show that the proposed attack is effective on this dataset too, achieving an attack success rate of over 50%.

Here we answer some general questions:

Q: Reproducibility in the real-world (Reviewer VvTE):

We did not reproduce physical experiments in the real world, due to ethical considerations, as observed by Reviewer D9vr.

Q: Main contribution/novelty (Reviewers D9vr, 2xxQ):

The novel contributions of the paper are 1) generation of physiologically plausible adversarial EEG signals by proposing a new method based on signals’ first derivative; 2) modeling the signal propagation over the human scalp based on its physical and electrical properties making the attack scenario more realistic. We believe that our algorithmic and conceptual contributions do not exist in prior work.

Q: Figure 1 in the Introduction (Reviewer VvTE, D9vr):

Previous related works in adversarial attacks in BCIs suppose a “jamming” module between the data acquisition source and the processing engine, assuming a stronger adversary that can access the EEG data during the data transmission (Zhang & Wu, 2019, Liu et al., 2021). In our work, we introduce a paradigm shift where an eventual attack can be delivered at the source of the data acquisition. This attack scenario is more realistic, even in the hardest scenario when the EEG signals are processed locally at the edge with embedded algorithms and not remotely transmitted, as the recent trend in edge computing dictates. This new scenario makes the previous attack models unrealistic. Figure 1 was meant to visually illustrate this paradigm shift that we introduce for the first time in an adversarial attack on BCIs.
To address a comment by Reviewer 2xxQ, Figure 1 is now moved to Appendix A in the updated submission.

Q: Why only EEGNet and not transformers or conventional signal processing methods (Reviewers 2xxQ, n45G)

We based our selection on the models which have been embedded on edge processors such as microcontrollers (Wang et al., 2018; Schneider et al., 2020) and selected EEGNet because it presents the best accuracy-energy trade-off making it the state-of-the-art embedded model for BCIs (Schneider et al., 2020).

We are aware of the usage of Transfomer models in the EEG domain. Even though they showed improved robustness against adversarial attacks in other domains, the baseline accuracy of those models on EEG is lower than EEGNet (Sun et al., 2021) and has not yet been demonstrated to be suitable for deployment on smart wearable BCI systems.

Common spatial patterns-based models have been also deployed on edge processors, such as FPGAs (Bewalfi et al., 2018; Malekmohammadi et al., 2019), however, compared to the EEGNet implementation by Schneider et al. (2020), they consume 893.2x and 2.9x more energy and have lower classification accuracy.
Recently, a Riemannian-based approach has also been deployed on microcontrollers, with an energy consumption that is 1.9x higher (Wang et al., 2021).

---

> ### Author Response · Authors · 2021-11-19
> **References for "Major additions and answer to common questions"**
>
> Kais Belwafi, Olivier Romain, Sofien Gannouni, Fakhreddine Ghaffari, Ridha Djemal, and Bouraoui Ouni. An embedded implementation based on adaptive filter bank for brain–computer interface systems. Journal of Neuroscience Methods, 2018. doi: 10.1016/j.jneumeth.2018.04.013.
>
> Xintian Han, Yuxuan Hu, Luca Foschini, Larry Chinitz, Lior Jankelson, and Rajesh Ranganath. Deep learning models for electrocardiograms are susceptible to adversarial attack. Nature medicine, 26 (3):360–363, 2020.
>
> Zihan Liu, Lubin Meng, Xiao Zhang, Weili Fang, and Dongrui Wu. Universal adversarial perturbations for CNN classifiers in EEG-based BCIs, 2021.
>
> Alireza Malekmohammadi, Hoda Mohammadzade, Alireza Chamanzar, Mahdi Shabany, and Benyamin Ghojogh. An efficient hardware implementation for a motor imagery brain computer interface system. Scientia Iranica, 26:72–94, 2019.
>
> Tibor Schneider, Xiaying Wang, Michael Hersche, Lukas Cavigelli, and Luca Benini. Q-eegnet: an energy-efficient 8-bit quantized parallel eegnet implementation for edge motor-imagery brain machine interfaces. In 2020 IEEE International Conference on Smart Computing (SMARTCOMP), pp. 284–289, 2020.
>
> Jiayao Sun, Jin Xie and Huihui Zhou. EEG Classification with transformer-based models. IEEE 3rd Global Conference on Life Sciences and Technologies (LifeTech), pp. 92-93, 2021.
>
> Xiaying Wang, Michael Hersche, Batuhan Tömekce, Burak Kaya, Michele Magno, and Luca Benini. An accurate eegnet-based motor-imagery brain–computer interface for low-power edge computing. In 2020 IEEE International Symposium on Medical Measurements and Applications (MeMeA), pp. 1–6, 2020.
>
> Xiaying Wang, Tibor Schneider, Michael Hersche, Lukas Cavigelli and Luca Benini. Mixed-Precision Quantization and Parallel Implementation of Multispectral Riemannian Classification for Brain-Machine Interfaces. IEEE International Symposium on Circuits and Systems (ISCAS), pp. 1-5, 2021.
>
> Xiao Zhang and Dongrui Wu. On the vulnerability of CNN classifiers in EEG-based BCIs. IEEE Transactions on Neural Systems and Rehabilitation Engineering, 27(5):814–825, 2019.

---

### Decision · Program_Chairs · 2022-01-20

**Decision:**

Reject

**Comment:**

This work has triggered engaged discussions between authors and reviewers and also among reviewers.
These conversations have highlighted the potential weaknesses of the contribution, namely that the work
is a proof-of-concept experimentally (although arguably for ethical reasons) and that the
overall theoretical contribution is not strong.

Despite the merit of this work, and given the strong expectations of ICLR, I don't feel that this work
can be endorsed for publication at ICLR 2022.